# Integrative genomics identifies a convergent molecular subtype that links epigenomic with transcriptomic differences in autism

Gokul Ramaswami[1], Hyejung Won [1,8], Michael J. Gandal [1,2,3,4], Jillian Haney[1,2], Jerry C. Wang [1], Chloe C. Y. Wong[5], Wenjie Sun[6], Shyam Prabhakar [6], Jonathan Mill [7] & Daniel H. Geschwind [1,3,4✉]

Autism spectrum disorder (ASD) is a phenotypically and genetically heterogeneous neuro-developmental disorder. Despite this heterogeneity, previous studies have shown patterns of molecular convergence in post-mortem brain tissue from autistic subjects. Here, we integrate genome-wide measures of mRNA expression, miRNA expression, DNA methylation, and histone acetylation from ASD and control brains to identify a convergent molecular subtype of ASD with shared dysregulation across both the epigenome and transcriptome. Focusing on this convergent subtype, we substantially expand the repertoire of differentially expressed genes in ASD and identify a component of upregulated immune processes that are associated with hypomethylation. We utilize eQTL and chromosome conformation datasets to link differentially acetylated regions with their cognate genes and identify an enrichment of ASD genetic risk variants in hyperacetylated noncoding regulatory regions linked to neuronal genes. These findings help elucidate how diverse genetic risk factors converge onto specific molecular processes in ASD.

[1] Program in Neurogenetics, Department of Neurology, David Geffen School of Medicine, University of California Los Angeles, Los Angeles, CA 90095, USA. [2] Department of Psychiatry, Semel Institute, David Geffen School of Medicine, University of California Los Angeles, Los Angeles, CA 90095, USA. [3] Center for Autism Research and Treatment, Semel Institute, David Geffen School of Medicine, University of California Los Angeles, Los Angeles, CA 90095, USA. [4] Department of Human Genetics, David Geffen School of Medicine, University of California Los Angeles, Los Angeles, CA 90095, USA. [5] Institute of Psychiatry, Psychology and Neuroscience, King's College London, De Crespigny Park, London SE5 8AB, UK. [6] Computational and Systems Biology, Genome Institute of Singapore, Singapore 138672, Singapore. [7] University of Exeter Medical School, University of Exeter, Exeter EX2 5DW, UK. [8] Present address: Department of Genetics and UNC Neuroscience Center, University of North Carolina, Chapel Hill, NC 27599, USA. ✉email: dhg@mednet.ucla.edu

Autism spectrum disorder (ASD) is a prevalent neurodevelopmental disorder characterized by impaired social interactions with repetitive and restrictive behaviors[1]. Although ASD is highly heritable, its genetic etiology is complex, with ~1000 risk genes implicated[2]. Assessment of ASD risk is challenging due to its genetic architecture which encompasses alleles of varying frequencies (common, rare, very rare) and inheritance patterns (Mendelian autosomal and X-linked, additive, de novo)[3–5] that likely interact together within individuals and families[6,7].

Surprisingly, despite this genetic complexity, molecular studies have identified consistent patterns of changes in post-mortem brain tissue from ASD subjects[8–12]. At the transcriptomic level, ASD brains exhibit downregulation of genes involved in neuronal activity with a concomitant upregulation of genes involved in microglial and astrocyte-mediated inflammation[9,13]. Additionally, there is a shared pattern of microRNA (miRNA) dysregulation directly targeting downregulated neuronal genes as well as upregulated astrocyte genes[12]. At the epigenomic level, ASD brains exhibit DNA methylation differences in genomic regions related to immunity and neuronal regulation[11,14]. Additionally, there are differences in histone acetylation (H3K27ac) associated with genes involved in synaptic transmission and morphogenesis[10]. To date, these molecular datasets have not been comprehensively integrated and analyzed together, which could provide a better understanding of how epigenetic changes directly regulate the expression of their cognate genes and how these processes are related. Additionally, despite evidence for shared patterns of molecular dysregulation, only approximately two-thirds of ASD brain samples exhibit this major shared molecular pattern, indicating the potential for distinct molecular subtypes. Such heterogeneity among ASD cases would also be expected to reduce power to identify disease-related signals, providing another rationale for the identification of subtypes.

Systems-level integration of multi-omic datasets has been a successful strategy to identify molecular subtypes and elucidate causal mechanisms in cancer[15,16]. However, it has not yet been applied to neurodevelopmental disorders, including ASD. In this study, we utilize similarity network fusion (SNF), an integrative method that has identified molecular subtypes when integrating transcriptomic with epigenomic datasets in cancer[17], to integrate mRNA expression, miRNA expression, DNA methylation, and histone acetylation datasets from ASD brain (Fig. 1a). This unbiased data-driven analysis identifies two distinct molecular subtypes of ASD, one, which represents the majority of cases, showing a cohesive molecular pattern, and the other without consistent changes in molecular measures. By analyzing ASD brains according to subtype, which significantly reduces heterogeneity, we are able to identify substantially more differentially expressed mRNA genes compared to previous analyses. We identify differentially expressed miRNAs, differentially methylated promoters and gene bodies, as well as differentially acetylated genomic regions and assess the extent to which these regulatory mechanisms influence gene expression in ASD. Finally, we find an enrichment of ASD genetic risk in regulatory regions linked to neuronal genes that are hyperacetylated in ASD brains, suggesting a causal role for these elements.

## Results
### Integration of multi-omic data from ASD and control brains.
We integrated previously published datasets on mRNA expression[9], miRNA expression[12], DNA methylation[11], and histone acetylation[10] from a cohort of 48 ASD and 45 control brains (Supplementary Data 1 and Fig. 1a). We only analyzed the samples originating from the frontal and temporal cortex, because

previous studies found ASD dysregulated features were predominantly localized to the cerebral cortex and substantially attenuated in the cerebellum. For mRNA and miRNA expression, we used normalized gene quantifications and differential expression summary statistics from the previous studies[9,12] ("Methods").

For DNA methylation, we used the normalized probe quantifications from the previous study[11] and collapsed probe-level measurements onto 21880 gene promoters and 24458 gene bodies to facilitate comparisons between methylation and expression ("Methods"). An initial differential methylation analysis identified 2578 and 1262 differentially methylated promoters and gene bodies, respectively, at an FDR < 10% ("Methods", Supplementary Fig. 1a, b, and Supplementary Data 4). The genes with differential promoter methylation were largely distinct from genes with differential gene body methylation (Supplementary Fig. 1e, f). However, the loadings for each sample along the first principal component of differential promoter and gene body methylation were almost identical (Supplementary Fig. 1d) and not correlated with any potential confounders (Supplementary Fig. 1c), suggesting a coherent regulatory mechanism.

For histone acetylation, we quantified the consensus H3K27ac peaks identified in the previous study[10] in all ASD and control samples including the samples marked as atypical in the previous study[10] ("Methods"). An initial differential acetylation analysis identified 2156 differentially acetylated regions at an FDR < 20% ("Methods", Supplementary Fig. 2a, and Supplementary Data 5). Although this was fewer differentially acetylated regions than previously identified (Supplementary Fig. 2b), we show that this depletion is an artifact of subtype heterogeneity in the ASD samples (see below "Subtype-specific histone acetylation differences in ASD" section).

### Identification of two ASD molecular subtypes.
In the previous molecular studies[9–12], we noticed that approximately two-thirds of ASD brain samples clustered together based on the differential signal for each dataset. To formally assess ASD molecular heterogeneity across the four different datasets, we used SNF[17] to integrate differential mRNA expression, miRNA expression, DNA methylation, and histone acetylation for 30 ASD and 17 control samples that were present in all 4 molecular datasets ("Methods"). SNF creates an integrative sample-sample similarity network by quantifying sample-sample relationships within each individual dataset and then integrating these sample-sample relationships across all of the datasets[17]. The clustering of sample relationships is a major advantage of SNF, as compared to alternative data integration methods that cluster gene relationships which can be sensitive to differing normalization methods between data types[18].

The sample loadings along the first principal component of each differential molecular level recapitulated known regulatory relationships, with differential acetylation ($R = 0.73$) and differential miRNA expression ($R = 0.51$) being highly correlated to differential mRNA expression, whereas differential methylation was less correlated with expression ($R = 0.13$)[19,20] (Fig. 1b). Using this similarity network, samples divided into two distinct clusters (Fig. 1c), one of which consisted entirely of ASD samples that loaded strongly onto the differential transcriptomic and epigenomic signatures (SNF Group 2). Therefore, we grouped these samples together as the ASD Convergent Subtype. The other cluster consisted of ASD samples that did not load onto the initial differential signatures and were indistinguishable from controls (SNF Group 1). Therefore, we grouped these samples together as the ASD Disparate Subtype. We built a logistic

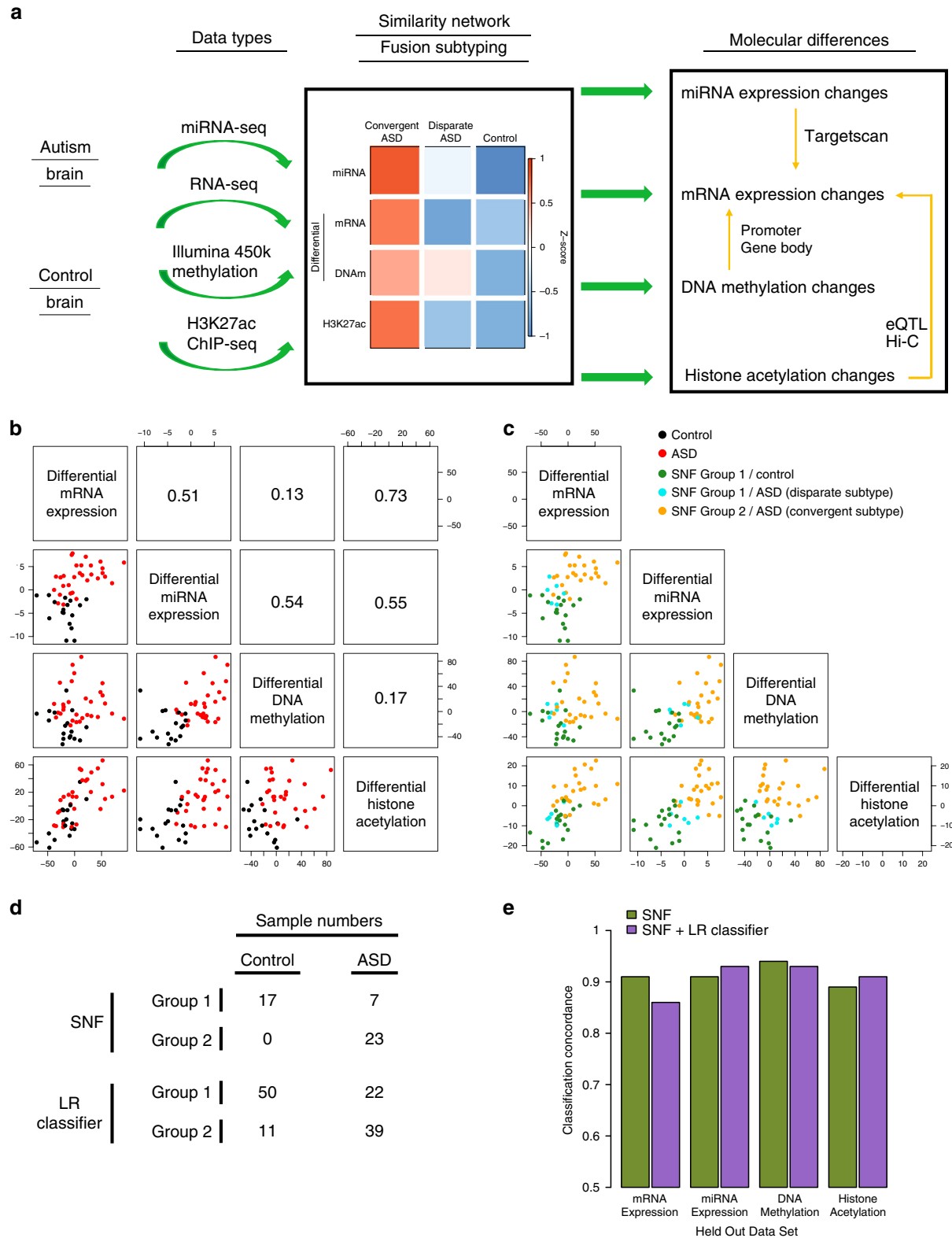

**Fig. 1 SNF to identify ASD molecular subtypes. a** Overview of data integration and molecular subtyping to characterize the cascade of molecular changes in ASD. **b** Relationship between sample loadings on the first principal component of differential mRNA expression, miRNA expression, DNA methylation, and histone acetylation. **c** Identification of two sample clusters using SNF: SNF Group 1 and SNF Group 2. ASD samples in SNF Group 1 constitute the Disparate Subtype, whereas ASD samples in SNF Group 2 constitute the Convergent Subtype. **d** Number of samples classified into the two cluster groups using SNF and logistic regression (LR) classifiers. **e** Comparison of SNF clustering and logistic regression (LR) classification assignments when utilizing three out of four datasets (see Supplementary Fig. 5). The concordance of sample assignments to those when using the complete dataset are plotted.

regression classifier to assign the 61 ASD and 61 control samples that were not used in at least one of the four classification datasets into either the Convergent or Disparate subtypes ("Methods", Fig. 1d and Supplementary Fig. 4). Interestingly, 11 of the 43 ASD individuals with samples from both frontal and temporal cortex were classified into different molecular subtypes in the two cortical regions (Supplementary Fig. 4d), a significant difference in comparison to only 1 of the 33 control individuals ($p = 0.0096$, Fisher's Exact Test). This finding is consistent with potential molecular heterogeneity across different cortical regions in ASD.

SNF subtype assignments were robust to clustering methodology and comprehensive leave one out cross validation (Supplementary Fig. 3c, d). We further tested clustering robustness by leaving out each dataset and performing SNF clustering and logistic regression classification using the remaining three datasets. We found the resultant sample subtype assignments were highly concordant with the subtype assignments identified using the entire four dataset collection (range = 0.86–0.93; Fig. 1e and Supplementary Fig. 5). Additionally, the ASD subtype assignments were not correlated with, or driven by, biological or technical covariates, including age, sex, RNA quality, cell fraction, and post mortem interval (Supplementary Fig. 6). Finally, to compare with the sample classifications above, which were generated using only differential features in each dataset, we attempted to cluster the samples with SNF using all features in the transcriptomic and epigenomic datasets. We find no clear separation between ASD and control samples (Supplementary Fig. 3a, b), demonstrating that molecular differences between ASD and control brains are restricted to specific differentiating features and are not a general genome-wide phenomenon.

**Subtype-specific mRNA expression differences in ASD.** To leverage the increased power of analyzing a more homogenous set of cases, we performed differential mRNA expression analyses separately for ASD Convergent and Disparate subtypes against control samples for each gene ("Methods" and Supplementary Fig. 7a, b). For the ASD Convergent subtype, we observed 5439 differentially expressed genes at an FDR < 5%, 2283 of which were upregulated and 3156 downregulated in ASD (Supplementary Data 2). We reproduce 94.5% of the differentially expressed genes from the previous study[9] and identify an additional 4356 genes at the same statistical threshold (Fig. 2a, b and Supplementary Fig. 7f), demonstrating the utility of this subtype-specific approach. The top gene ontology enrichments are very similar to those previously identified, showing an upregulation of genes involved in immune response and a downregulation of genes involved in synaptic transmission and neuronal ion transport (Supplementary Fig. 7c, d). In contrast, for the ASD Disparate subtype, we found no differentially expressed genes.

Next, we identified mRNA co-expression modules that were differentially associated between ASD and control individuals in the cortical co-expression network defined previously[9]. For each gene module, we tested whether the two ASD subtypes and control samples had differences in their association with the module eigengene, a summary measure of module expression level (Supplementary Fig. 7e). For the ASD Convergent subtype, we found 13 differentially associated co-expression modules at an FDR < 5%, including the 6 ASD-associated modules identified previously[9] and 7 ASD-associated modules additionally identified in this study (Fig. 2c). In contrast, for the ASD Disparate subtype, we did not find any differentially associated co-expression modules.

Of the seven additionally identified ASD-associated co-expression modules in this study, four were downregulated in ASD: mRNA.M1, a module representing neurogenesis, mRNA.

M3, a module representing mitochondrial function in neurons, which has been previously implicated in ASD[21], mRNA.M7, a module with no functional enrichments, and mRNA.M17, a module representing synaptic signaling and vesicle transport in neurons (Fig. 2d, e). To gain insight into neuronal downregulation in ASD at a finer resolution, we compared ASD downregulated modules to neuronal cell type-specific markers identified from single-nuclei RNA sequencing of post-mortem human cortex[22]. ASD downregulated modules are significantly enriched with markers of both inhibitory and excitatory neurons (Fig. 2f). The strongest enrichments are inhibitory neuron subtypes expressing *SST* or *PVALB*, derived from the medial ganglionic eminence, as well as deep layer excitatory neurons expressing *RORB* or *FEZF2* (Fig. 2f), suggesting that the number and/or activity of these cells is decreased in ASD, consistent with a recent single-cell analysis of post mortem ASD brain[23].

Three of the additionally identified ASD-associated modules were upregulated in ASD: mRNA.M15, a module representing metabolic processes and transcriptional regulation in glia, mRNA.M21, a module representing ribosomal translational, and mRNA.M23, a module enriched with astrocyte markers. Module mRNA.M15 (Fig. 2g, h) was particularly interesting, because one of its top hub genes is *REST*, a transcriptional repressor with critical roles in repressing neural genes in non-neural cells[24]. Although module mRNA.M15 is enriched with microglial markers (Supplementary Fig. 7g), it exhibits a markedly different transcriptional profile[25] than the previously identified ASD upregulated microglial module mRNA.M19 (Fig. 2i). Module mRNA.M19 is specifically enriched with genes marking microglial activation[26], suggesting that it is directly related to neural-immune response. In contrast, module mRNA.M15 is enriched with markers of juvenile or aging glia, suggesting it may be related to glial growth and maturation. In general, cellular processes underlying broad categories of immuno-glial cell types are upregulated in ASD.

**Subtype-specific miRNA expression differences in ASD.** We conducted differential miRNA expression analyses for ASD Convergent and Disparate subtypes against control samples for each mature miRNA transcript (Methods and Supplementary Fig. 8a, b). For the ASD Convergent subtype, we identified 43 differentially expressed miRNAs at an FDR < 5%, 28 upregulated and 15 downregulated in ASD that highly overlapped with previous work (Supplementary Fig. 8c, d and Supplementary Data 3)[12]. We analyzed differentially associated miRNA co-expression modules in the miRNA co-expression network defined previously[12]. For the ASD Convergent subtype, we found the same three miRNA co-expression modules differentially associated at an FDR < 5% as the previous study: miRNA.brown, which is downregulated in ASD, as well as miRNA.magenta and miRNA.yellow, which are upregulated (Supplementary Fig. 8e). We used TargetScan[27] to predict mRNA targets for the top hubs of each differentially associated miRNA co-expression module (Supplementary Data 3). We found an enrichment of genes in the upregulated mRNA.M19 module within the predicted targets of the miRNA.brown module (Supplementary Fig. 8f), suggesting the downregulation of these miRNAs may contribute to the upregulation of immune processes in ASD. We also found an enrichment of genes in the downregulated mRNA.M16 and mRNA.M3 modules within the predicted targets of miRNA.magenta and miRNA.yellow (Supplementary Fig. 8f), suggesting the upregulation of these miRNAs may contribute to the downregulation of neuronal processes in ASD. In contrast, for the ASD Disparate subtype, we did not find any differentially expressed miRNAs or differentially associated miRNA co-expression

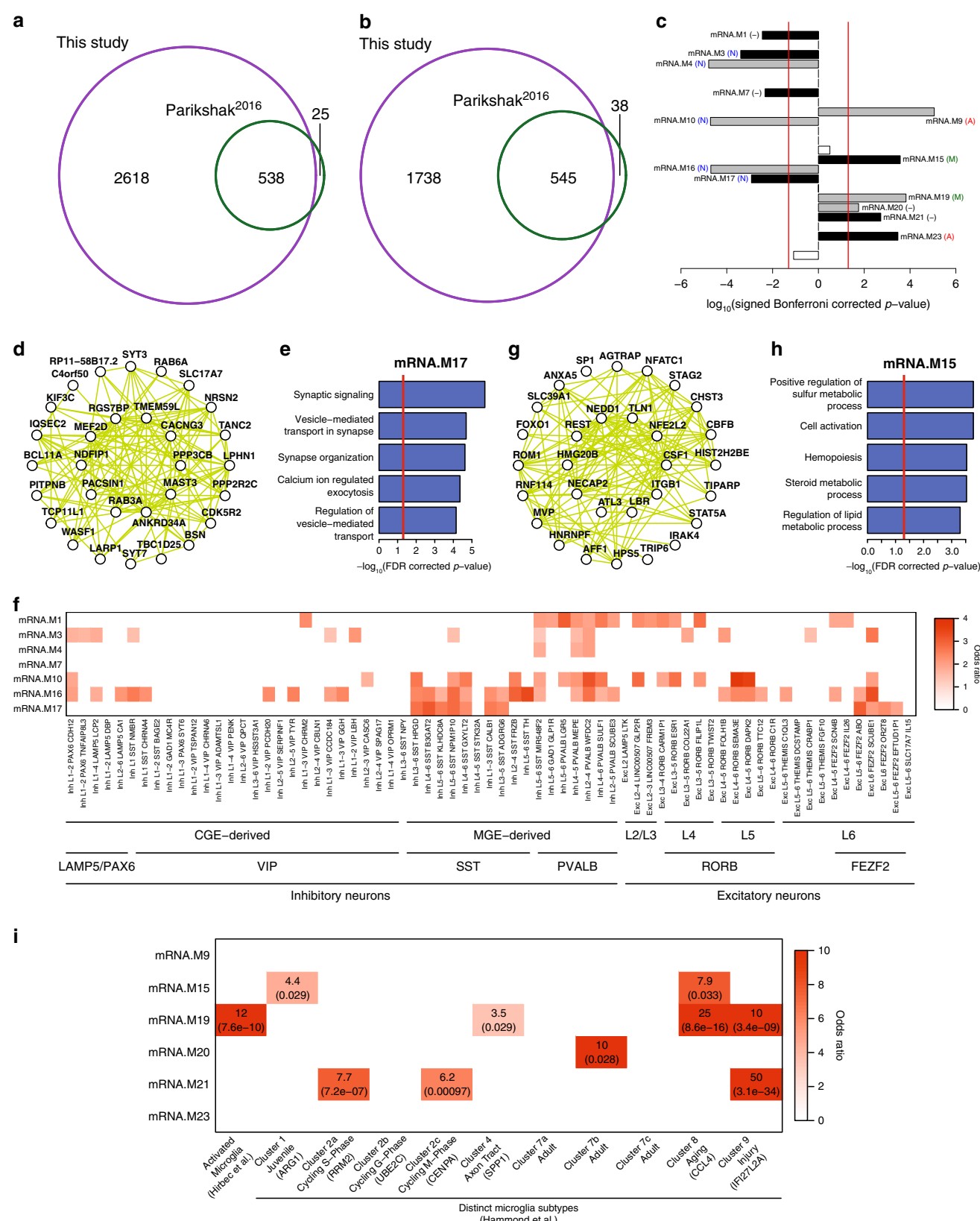

modules. Overall, subtype-specific analyses of miRNA expression largely recapitulated findings from the previous work[12].

**Subtype-specific DNA methylation differences in ASD.** We conducted differential DNA methylation analyses for ASD Convergent and Disparate subtypes against control samples for gene promoters and gene bodies ("Methods"). For the ASD Convergent subtype, we identified 3013 differentially methylated gene promoters at an FDR < 5%, 2298 hypermethylated and 715 hypomethylated in ASD (Supplementary Fig. 9b and Supplementary Data 4). ASD hypermethylated gene promoters are enriched in RNA processing genes, while ASD hypomethylated

**Fig. 2 mRNA expression differences in ASD. a** Overlap in ASD downregulated genes identified in this study with Parikshak et al.[9]. **b** Overlap in ASD upregulated genes identified in this study with Parikshak et al.[9]. **c** Signed association of mRNA co-expression module eigengenes with diagnosis (Bonferroni-corrected p-value from a linear mixed effects model, see Supplementary Fig. 7e). Positive values indicate modules with an increased expression in ASD samples. Gray and black bars with labels signify ASD-associated modules identified in Parikshak et al., and those additionally identified in this study, respectively. Cell type enrichment for each module is shown in parenthesis: neuron (N), astrocyte (A), microglia (M), and no enrichment (−) (see Supplementary Fig. 7g). **d** Top 30 hub genes and 300 connections for co-expression module mRNA.M17. **e** Top gene ontology enrichments for co-expression module mRNA.M17. Ontology enrichments were calculated by g:Profiler with FDR corrected p-values. **f** Enrichment of ASD downregulated neuronal co-expression modules with neuronal cell-type markers identified from single-nuclei RNA sequencing[22]. Enrichments were calculated using a logistic regression model and p-values were adjusted for multiple testing using FDR correction. Only those enrichments with odds ratio >1 and FDR corrected p-value < 0.05 are shown. **g** Top 30 hub genes and 300 connections for co-expression module mRNA.M15. **h** Top gene ontology enrichments for co-expression module mRNA.M15. Ontology enrichments were calculated by g:Profiler with FDR corrected p-values. **i** Enrichment of ASD upregulated glial co-expression modules with microglial activated genes[26] and microglial cell-type markers[25]. Enrichments were calculated using a logistic regression model and p-values, which are shown in parentheses, were adjusted for multiple testing using FDR correction. Only those enrichments with odds ratio >1 and FDR corrected p-value < 0.05 are shown.

gene promoters are enriched in chemical sensory receptor genes (Supplementary Fig. 9d, e). We identified 1460 differentially methylated gene bodies at an FDR < 5%, 678 hypermethylated and 782 hypomethylated in ASD (Supplementary Fig. 10b and Supplementary Data 4). ASD hypermethylated gene bodies are enriched in RNA processing genes, while ASD hypomethylated gene bodies are enriched in keratinization and bile acid transport genes (Supplementary Fig. 10d, e). We assigned the previously identified differentially methylated probes[11] to their corresponding promoter or gene body annotation and now identify fifty and eleven-fold more differentially methylated promoters (Supplementary Fig. 9h, i) and gene bodies (Supplementary Fig. 10h, i), respectively. In contrast, for the ASD Disparate subtype, we did not find any differentially methylated gene promoters or gene bodies (Supplementary Figs. 9c and 10c). Genes with differentially methylated gene promoters were largely distinct from those with differentially methylated gene bodies although there was a greater overlap in hypermethylated genes due to their shared biological enrichments (Fig. 3a, b). There were 596 and 305 genes that were both differentially expressed as well as differentially methylated at gene promoters and gene bodies, respectively. As expected, there was a negative correlation between differential expression and differential methylation for gene promoters (Fig. 3c). Surprisingly, there was also a negative correlation with gene body methylation (Fig. 3d), suggesting that in the context of ASD, gene body methylation is associated with negative regulation of gene expression, or reflects a secondary response.

We generated co-methylation networks for both promoters and gene bodies (Supplementary Figs. 9a and 10a), which recapitulated many aspects of the probe-level co-methylation network that was previously built[11] (Supplementary Fig. 11a, b). We identified eight ASD-associated promoter co-methylation modules, four hypermethylated and four hypomethylated in ASD (FDR < 0.05; Supplementary Fig. 9f). The hypermethylated promoter modules were: Prom.midnightblue, representing coenzyme A biosynthesis, Prom.pink, a module with no functional enrichments, Prom.tan, enriched with oligodendrocyte cell markers, and Prom.turquoise, enriched with astrocyte cell markers and representing RNA processing. The hypomethylated modules were: Prom.brown and Prom.greenyellow, two modules representing sensory perception, as well as Prom.lightcyan and Prom.lightgreen, two modules representing immune processes.

At the gene body level, we identified 10 ASD-associated co-methylation modules, 4 hypermethylated and 6 hypomethylated in ASD (FDR < 0.05; Supplementary Fig. 10f). The hypermethylated modules were: GB.blue, representing RNA processing, GB.cyan, enriched in neuron cell markers and representing the unfolded protein response, GB.darkred, enriched in neuron cell

markers and representing mitochondrial activity, and GB.royalblue, a module with no functional enrichments. The hypomethylated modules were: GB.black, GB.darkgreen, GB.lightcyan, and GB.salmon, 4 modules representing immune processes, as well as GB.green, a module representing glucuronidation, and GB.yellow, representing bile acid ion transport. In contrast, for the ASD Disparate subtype, we did not find any ASD-associated promoter or gene body co-methylation modules.

Overall, the ASD-associated co-methylation modules did not show significant global overlap with the ASD-associated co-expression modules at the gene level (Supplementary Fig. 11c, d), suggesting that differential methylation is not a prominent driver of differential gene expression. The largest overlaps are between hypomethylated co-methylation modules and upregulated co-expression modules involved in immune processes. In particular, Prom.lightgreen overlaps significantly with mRNA.M19 and GB.darkgreen overlaps significantly with mRNA.M15, suggesting that the ASD-associated upregulation in immune activation is, in part, regulated by a decrease in DNA methylation at promoters and gene bodies (Fig. 3e–j).

**Subtype-specific histone acetylation differences in ASD**. We next performed differential histone acetylation analyses for ASD Convergent and Disparate subtypes against control samples for H3K27ac peaks in the genome ("Methods"). For the ASD Convergent subtype, we identified 15967 differentially acetylated peaks at an FDR < 10%, 8707 hyperacetylated and 7260 hypoacetylated in ASD (Supplementary Fig. 12a and Supplementary Data 5). There was a strong overlap with differentially acetylated peaks identified in a previous study (Supplementary Fig. 12c, d)[10]. In contrast, for the ASD Disparate subtype, we did not find any differentially acetylated peaks (Supplementary Fig. 12b). Using GREAT to assess ontology enrichments for genes closest to each differentially acetylated peak[28], we find that genes proximal to ASD hyperacetylated peaks are enriched in synaptic potential, while genes proximal to ASD hypoacetylated peaks are enriched in neurogenesis and organ development (Fig. 4a, b). H3K27ac is known to mark active promoters[29], and as expected, differentially acetylated peaks in gene promoters were strongly positively correlated ($R = 0.25$) with differential expression (Fig. 4d). Surprisingly, we found that the relationship between differential promoter acetylation and differential expression was cell type-specific, with hyperacetylated promoters associated with upregulated microglial genes and downregulated neuronal genes, whereas hypoacetylated promoters were associated with upregulated astrocyte genes and downregulated oligodendrocyte genes (Fig. 4e).

In addition to marking promoters, H3K27ac also marks distal enhancers up to 1 MB away from gene transcription start sites (TSS)[30]. We utilized expression quantitative trait loci (eQTL) and

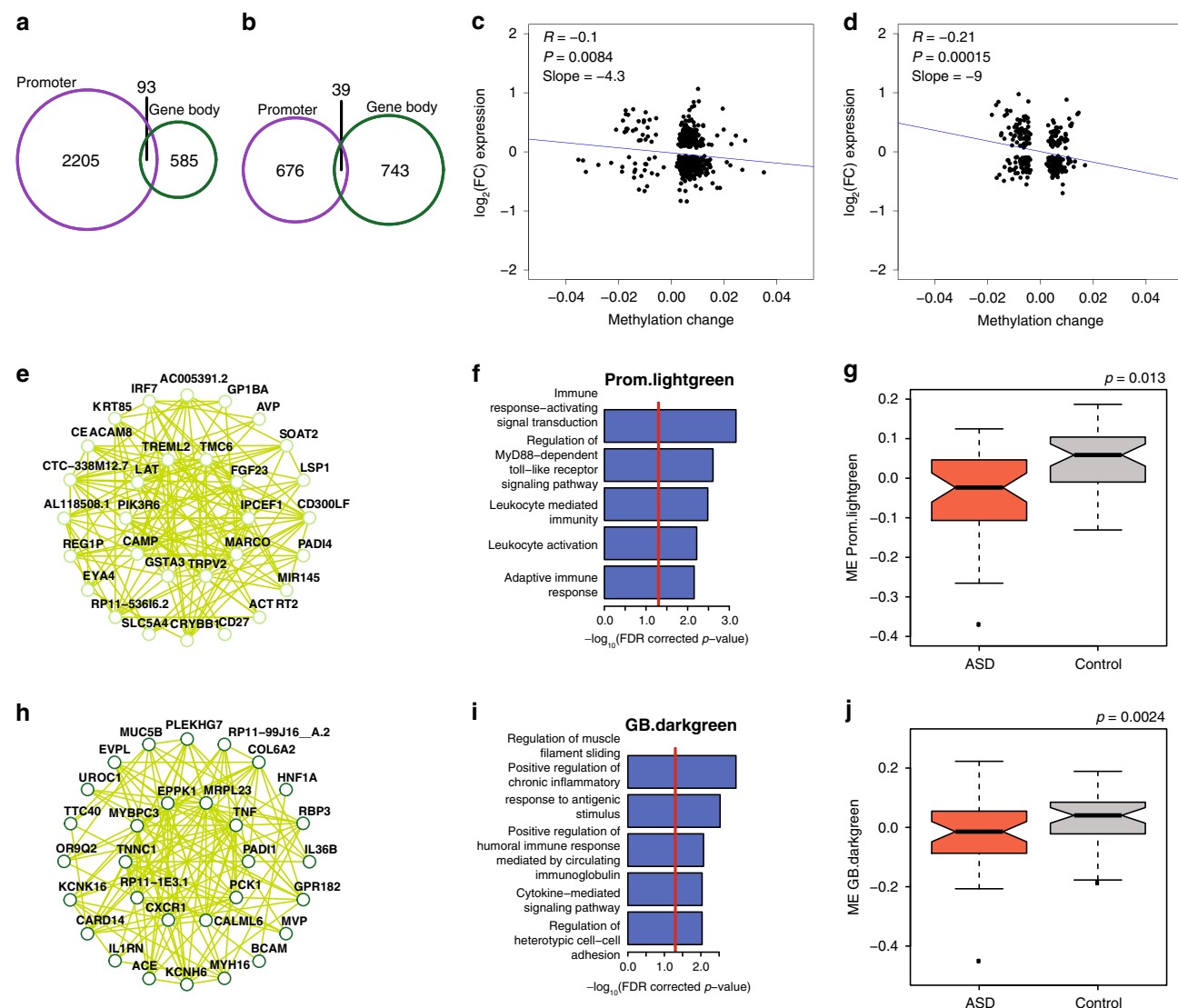

**Fig. 3 DNA methylation differences in ASD. a** Overlap in ASD hypermethylated gene promoters and gene bodies. **b** Overlap in ASD hypomethylated gene promoters and gene bodies. **c** Correlation between expression and methylation changes for genes that have differential promoter methylation and are differentially expressed. A linear model was used to correlate differential expression with differential methylation. *P*-value is from fit of linear model. **d** Correlation between expression and methylation changes for genes that have differential gene body methylation and are differentially expressed. A linear model was used to correlate differential expression with differential methylation. *P*-value is from fit of linear model. **e** Top 30 hub genes and 300 connections for promoter co-methylation module Prom.lightgreen. **f** Top gene ontology enrichments for promoter co-methylation module Prom.lightgreen. Ontology enrichments were calculated by g:Profiler with FDR corrected *p*-values. **g** Promoter co-methylation module Prom.lightgreen eigengene values for ASD and control samples. *P*-value is from fit of a linear mixed effects model (see Supplementary Fig. 9f). **h** Top 30 hub genes and 300 connections for gene body co-methylation module GB.darkgreen. **i** Top gene ontology enrichments for gene body co-methylation module GB.darkgreen. Ontology enrichments were calculated by g:Profiler with FDR corrected *p*-values. **J** Gene body co-methylation module GB.darkgreen eigengene values for ASD and control samples. *P*-value is from fit of a linear mixed effects model (see Supplementary Fig. 10f). For boxplots in **g**, **j**, the center of the box is the median value, the bounds of the box are the 75th and 25th percentile values, the whiskers extend out from the box to 1.5 times the interquartile range of the box, and outlier values are presented as individual points.

chromatin conformation capture (Hi-C) datasets from bulk adult brain tissue[31], as well as Hi-C data using sorted neuronal and glial cells from adult brain tissue ("Methods") to link distal differential H3K27ac peaks with their cognate genes (Fig. 4c; "Methods"). As a baseline, we analyzed all differentially acetylated peaks within 1 MB of a differentially expressed gene TSS and found that the overall correlation between differential acetylation and differential expression was minute ($R = 0.018$, $p = 2e-8$) (Supplementary Fig. 12e). The correlations improved when linking differentially acetylated regions to differentially expressed genes using eQTL ($R = 0.084$, $p = 2.3e-9$) (Supplementary Fig. 12f), bulk Hi-C ($R = 0.075$, $p =$

1.4e-5) (Supplementary Fig. 12g), neuronal Hi-C ($R = 0.061$, $p = 7.6e-5$) (Supplementary Fig. 12h), and glial Hi-C ($R = 0.11$, $p = 1.4e-11$) (Supplementary Fig. 12i) datasets. The correlations were further improved when H3K27ac peaks were linked to genes using a combination of eQTL and bulk Hi-C ($R = 0.18$, $p = 9.2e-4$) (Supplementary Fig. 12j) or glial Hi-C ($R = 0.23$, $p = 2.9e-5$) (Supplementary Fig. 12l) datasets. However, the combination of eQTL and neuronal Hi-C datasets was uncorrelated ($R = 0$, $p = 0.36$) (Supplementary Fig. 12k), possibly reflecting the bias of bulk eQTL linkages to glial cells due to the greater number of glia as compared with neurons in the cerebral cortex[32].

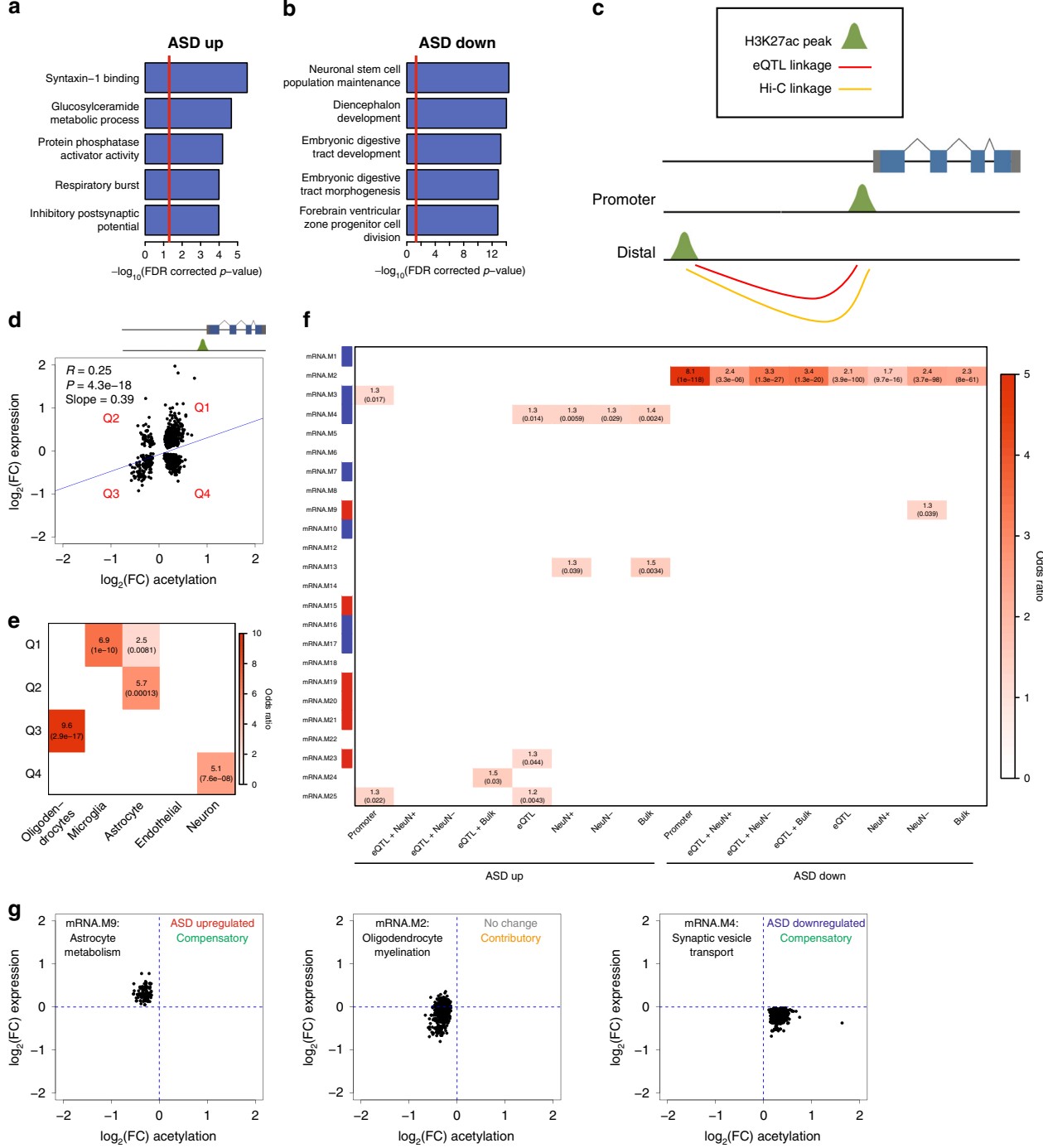

**Fig. 4 Histone acetylation differences in ASD. a** Top gene ontology enrichments when linking ASD hyperacetylated regions to proximal genes using GREAT[28]. *P*-values were adjusted for multiple testing by FDR correction. **b** Top gene ontology enrichments when linking ASD hypoacetylated regions to proximal genes using GREAT[28]. *P*-values were adjusted for multiple testing by FDR correction. **c** Schema to link H3K27ac regions with their cognate genes. H3K27ac peaks within promoters were directly assigned to the proximal gene. Distal H3K27ac peaks were assigned to genes using eQTL and Hi-C datasets. **d** Correlation between expression and acetylation changes for genes that have a differentially acetylated region within their promoter and are differentially expressed. *P*-value is from a linear model used to correlate differential expression with differential acetylation. The four separate quadrants are marked. **e** Cell type enrichments for the four quadrants in **d**. Enrichments were calculated using a logistic regression model and *p*-values, which are shown in parentheses, were adjusted for multiple testing using FDR correction. Only those enrichments with odds ratio >1 and FDR corrected *p*-value < 0.05 are shown. **f** Enrichment of cognate genes linked to differentially acetylated regions within mRNA co-expression modules. Modules with a significant relationship to diagnosis are marked along the *y* axes (red: increased expression in ASD; blue: decreased expression in ASD). Enrichments were calculated using a logistic regression model and *p*-values, which are shown in parentheses, were adjusted for multiple testing using FDR correction. Only those enrichments with odds ratio >1 and FDR corrected *p*-value < 0.05 are shown. **g** Relationship between expression and acetylation changes for differentially acetylated peaks linked to gene co-expression modules. The functional annotation for each module is represented in the top left corner. The association of each module to ASD diagnosis is represented in the top right corner as well as whether acetylation changes are contributory or compensatory to changes in expression.

We provide a listing of potential cognate genes linked to differentially acetylated peaks using the eQTL, neuronal Hi-C, glial Hi-C, and bulk Hi-C linkages (Supplementary Data 5). Using this list of cognate genes, we identified gene co-expression modules potentially regulated by ASD-associated acetylation changes (Fig. 4f). We find an enrichment of ASD hypoacetylated elements in mRNA.M2, an oligodendrocyte module harboring many ASD downregulated genes, suggesting that hypoacetylation is contributory to oligodendrocyte downregulation in ASD. Surprisingly, we find an enrichment of ASD hypoacetylated elements in mRNA.M9, an astrocyte module that is upregulated in ASD, and hyperacetylated elements in mRNA.M4, a neuron module that is downregulated in ASD. In both of these cases, acetylation changes are negatively correlated with expression changes. This suggests, for genes in these modules, that ASD-associated acetylation changes are not causal, but compensatory to gene expression changes (Fig. 4g).

**Enrichments of ASD heritability in dysregulated genomic features**. Differential gene expression or epigenetic changes may either be contributory to, or a consequence of disease. To provide a causal anchor, we used stratified LD score regression[33] to partition heritability of ASD risk variants from genome-wide association studies[4,34] into regions of the genome that are differentially expressed, methylated, or acetylated. We included inflammatory bowel disease (IBD) as comparator GWAS dataset[35], because IBD is a disorder of immune dysregulation, but it does not directly affect the brain. We also included Alzheimer's Disease (AD)[36], because AD is a neurological disorder, but like IBD, has a distinct genetic profile to ASD. We found a significant enrichment of ASD and IBD heritability in genomic regions that are hyperacetylated in ASD brains (Fig. 5a). Specifically, these enrichments are found at distal enhancer regions and not at gene promoters (Fig. 5b), highlighting the importance of noncoding regulatory elements. The IBD enrichment is likely marking hyperacetylated elements linked to microglial genes (Fig. 4d, e). Cognate genes linked to hyperacetylated enhancers are enriched within module mRNA.M4 (Fig. 4f), an ASD-downregulated neuronal module representing genes involved in synaptic vesicle transport (Fig. 5c, d). This finding supports previous observations that common genetic risk variants for ASD are enriched in regulatory regions of neuronal genes[4,13]. Among the genes within module mRNA.M4, six of its top hub genes are linked to elements hyperacetylated in ASD, including *DMTN* and *STX1B* (Fig. 5e, f). The enrichment of causal risk variants and subsequent downregulation of these hub genes within the module suggests that an increase in acetylation is likely an attempt to compensate for downregulation of key driver genes of this module.

Next, we analyzed whether there were ASD heritability enrichments in the co-expression and co-methylation network modules (Supplementary Fig. 13a–c). We found a significant enrichment of heritability within the promoter co-methylation module Prom.green, which represents genes involved in neurogenesis (Supplementary Fig. 13d, e), further strengthening the observation that ASD risk variants reside within regulatory regions of genes involved in neuronal function and neurogenesis.

Taken together (Fig. 6), our findings support a model whereby ASD risk variants perturb regulatory elements controlling the expression of neuronal genes, leading to the overall downregulation of synaptic signaling and neuronal ion transport. This in turn leads to the transcriptional upregulation of astrocyte and microglial mediated neural immune processes through a concomitant decrease in DNA methylation, decrease in expression of associated miRNAs, and increase of histone acetylation at regulatory elements of microglial genes. Finally, as a response to compensate these transcriptional changes, there is a decrease of histone acetylation at the promoters of upregulated astrocyte genes, and an increase of histone acetylation linked to down-regulated neuronal genes at the same regulatory elements initially impacted by the casual genetic variants.

## Discussion

We integrate mRNA expression, miRNA expression, DNA methylation, and histone acetylation datasets to identify a subtype of ASD brain samples with convergent dysregulation across the epigenome and transcriptome. By focusing on this convergent ASD subtype, we identify a four-fold expansion in differentially expressed mRNAs and co-expression modules encompassing the major processes of neuronal downregulation and immune upregulation. We identify thousands of differentially methylated gene promoters and gene bodies, but only a small proportion of the methylation changes, specifically hypomethylation of immune genes, seem to influence gene expression regulation. In contrast, histone acetylation is a strong positive regulator of mRNA expression[19] and we identify thousands of differentially acetylated regions across the genome and furthermore assign them to cognate genes using eQTL and chromosome conformation datasets. We find differentially acetylated regions enriched within dysregulated astrocyte and neuronal co-expression modules. Surprisingly the acetylation changes in these regions are negatively correlated with expression changes, implying that these changes are most likely compensatory.

Over 50% of ASD genetic liability is carried by small effect size common variants that are mostly noncoding[3]. One of the major challenges in characterizing these common variants is the ability to link noncoding regulatory regions with their cognate genes, which are often dynamically regulated across different cell types and developmental stages. In this study, we find an enrichment of ASD genetic risk within hyperacetylated regions of the genome, specifically those linked to downregulated neuronal genes. This is consistent with a previous study, which found enrichment of ASD risk variants in a down-regulated, neuronal-associated module in ASD post-mortem brain[13]. Moreover, whereas enrichment of ASD risk variants in brain regulatory elements, including those marked by H3K27ac, has been observed previously[4,37], we observe that ASD risk variants are specifically enriched in differentially regulated enhancers, providing a potential mechanistic understanding of how non-coding ASD risk variants likely impact gene regulation.

A major unanswered question is what molecular processes are involved in the ASD Disparate subtype samples. We were unable to find any consensus molecular dysregulation in these samples across all the datasets (Supplementary Figs. 7b, 8b, 9c, 10c, and 12b). We also assessed whether they were misdiagnosed. The available clinical records for these individuals are sparse, but we found no evidence to support misdiagnosis, nor did we find samples exhibiting differential expression signatures for four other neuropsychiatric disorders (Supplementary Fig. 14). We note that in 11 of the ASD subjects, the two different cortical regions were classified into different molecular subtypes (Supplementary Fig. 4d), suggesting that the extent of molecular dysregulation may vary across regions of the cortex in these individuals. As an initial assessment of regional heterogeneity, we analyzed RNA-seq data derived from four additional cortical areas in a subset of individuals in this study[21], including portions of the parietal and occipital lobes, and find some evidence of quantitative differences in gene expression dysregulation across the cortex (Supplementary Fig. 15). Future studies of ASD post-mortem brain samples with larger sample sizes, assessing more brain regions, and single-cell resolution, coupled to the

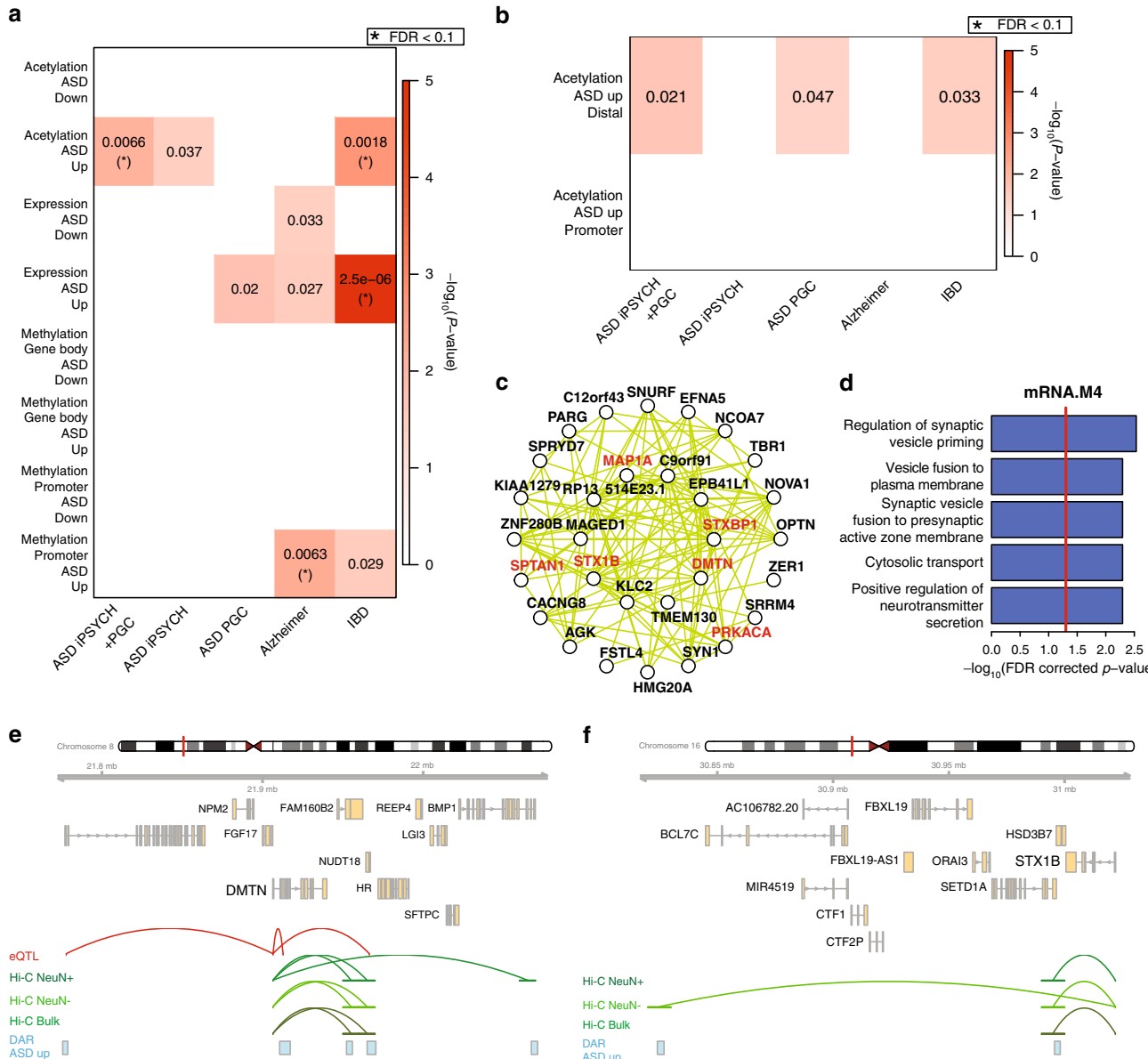

**Fig. 5 ASD genetic risk variant enrichments. a** Partitioned heritability enrichments for ASD, Alzheimer's, and IBD GWAS in differentially expressed, methylated, or acetylated regions of the genome. Uncorrected *p*-values < 0.05 are shown. *P*-values were also adjusted for multiple testing by FDR correction and those enrichments with adjusted *p*-value < 0.1 are marked with asterisks. **b** Partitioned heritability enrichments for ASD, Alzheimer's, and IBD GWAS in differentially acetylated regions of the genome within, or distal to, gene promoters. Uncorrected *p*-values < 0.05 are shown. *P*-values were also adjusted for multiple testing by FDR correction and those enrichments with adjusted *p*-value < 0.1 are marked with asterisks. **c** Top 30 hub genes and 300 connections for co-expression module mRNA.M4. **d** Top gene ontology enrichments for co-expression module mRNA.M4. Ontology enrichments were calculated by g:Profiler with FDR corrected *p*-values. **e**, **f** Genomic region around *DMTN* **e** and *STX1B* **f**. ASD-associated hyperacetylated regions are shown along with eQTL and Hi-C linkages to the gene TSS.

availability of comprehensive phenotypic data will be needed to further characterize the regional and cellular specificity of molecular changes in ASD.

## Methods

**Initial processing of datasets**. All molecular datasets came from a cohort of 48 ASD and 45 control brains from the Harvard Autism Tissue Program [https://hbtrc.mclean.harvard.edu/] and NIH Neuro Brain Bank [http://www.medschool.umaryland.edu/btbank/]. Initial analyses from the transcriptomic and epigenomic datasets have been previously published[9–12]. We restricted this study to only those samples originating from the frontal or temporal cortex. In the epigenomic studies[10,11], samples from the Oxford and MRC London Brain Banks were also present, however, we removed these samples because we did not have transcriptomic data for them.

For mRNA expression, we used the quantification of log2 RPKM values for 16310 genes in 82 ASD samples and 74 control samples from 47 ASD and 44 control brains from Parikshak et al.[9]. These RPKM values were normalized for gene length and GC content using CQN[38], but not adjusted for technical or biological covariates. We downloaded the differential mRNA gene expression summary statistics and mRNA co-expression network module definitions. We calculated principal components (SeqStatPCs) to summarize the following sequencing statistics: log10(TotalReads.picard), log10(Aligned.Reads.picard), log10(HQ.Aligned.Reads.picard), log10(PF.All.Bases.picard), log10(Coding.Bases.picard), log10(UTR.Bases.picard), log10(Intronic.Bases.picard), log10(Intergenic.bases.picard), Median.CV.Coverage.picard, Median.5prime.Bias.picard, Median.3prime.Bias.picard, Median.5to3prime.Bias.picard, AT.Dropout.picard, GC.Dropout.picard, and PropExonicReads.HTSC.

For miRNA expression, we used the quantification of log2 read counts mapping to 699 mature miRNAs in 60 ASD samples and 42 control samples from 39 ASD and 28 control brains from Wu et al.[12]. To balance the case/control cohorts with

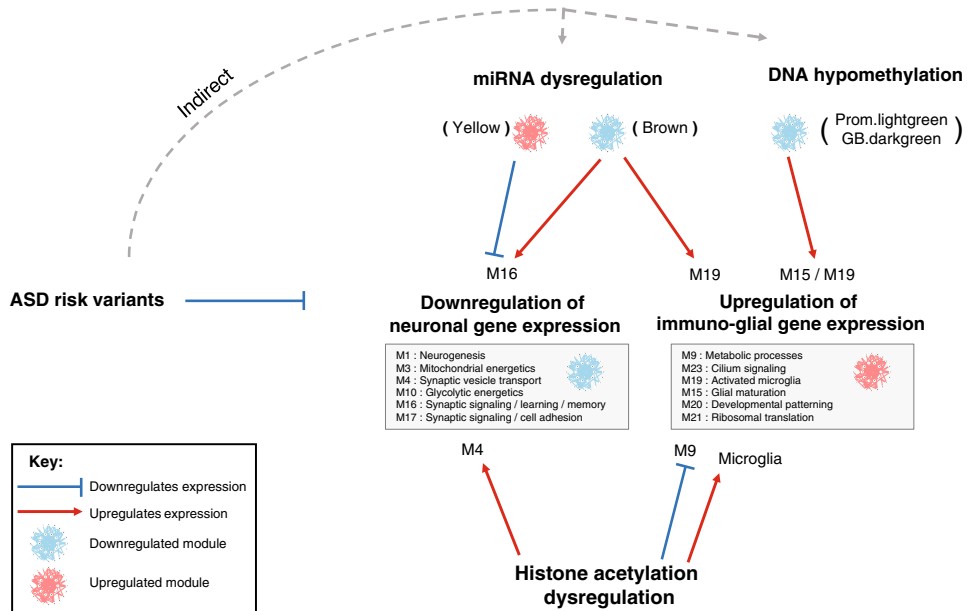

**Fig. 6 Schematic model of molecular dysregulation in ASD.** Integration across genetic, transcriptomic, and epigenomic levels in ASD. ASD risk variants primarily act with respect to neuronal genes, which are broadly down-regulated, likely indirectly, including via micro-RNA (e.g. yellow module). Other features, including another microRNA module (brown), and DNA hypomethylation are predicted to be compensatory or secondary and are associated with up-regulation of glial-immune genes. Histone acetylation patterns show a complex relationship, with some predicted to reflect an attempt to compensate for changes in gene expression (e.g. M4), while others are predicted to be likely driving the changes (microglia). Blue bar-headed and red arrows correspond to regulatory mechanisms predicted to decrease and increase gene expression, respectively.

respect to age, we removed all samples from younger individuals (Age ≤10) leaving us with 49 ASD samples and 42 control samples from 31 ASD and 28 control brains. These read counts were normalized for mature miRNA GC content using CQN[38] and batch effects using ComBat[39], but not adjusted for technical or biological covariates. We downloaded the differential miRNA expression summary statistics and miRNA co-expression network module definitions.

For DNA methylation, we used the quantification of methylation beta values for 417460 CpG probes across the genome in 74 ASD samples and 42 control brains from 42 ASD and 27 control brains from Wong et al.[11]. To balance the case/control cohorts with respect to age, we removed all samples from younger individuals (Age ≤10) leaving us with 56 ASD samples and 41 control samples from 33 ASD and 26 control brains. Probe quantifications were normalized using wateRmelon[40]. For each sample, the CET score was calculated[41], which is the proportion of neuronal vs glial cells. For samples where we had expression and/or acetylation data but did not have methylation data, we assigned their CET score as the average CET score. We collapsed the probe-level measurements to gene promoters and gene bodies by taking the average methylation level of all probes mapping onto gene promoters (2KB upstream of TSS to TSS) and gene bodies (TSS to transcription end site (TES)) using gencode v19 annotations[42]. We only kept gene promoters or gene bodies that contained 2 or more CpG probes. We downloaded the list of differentially methylated probes and co-methylation network identified from the cross-cortex analysis of Wong et al.[11]. These probes were assigned to gene promoters and gene bodies as described above.

For histone acetylation, we downloaded fastq files for 257 H3K27ac ChIP-seq samples as well as input samples from frontal cortex, temporal cortex, and cerebellum from Sun et al.[10] (https://www.synapse.org/#!Synapse:syn8104916). We mapped reads from each sample onto the hg19 reference genome using BWA-MEM[43] with default parameters. We removed duplicate reads using Picard tools [http://broadinstitute.github.io/picard/]. We utilized 56503 consensus H3K27ac peaks in the genome from Sun et al.[10] and quantified the levels of each consensus peak in each sample by counting the number of overlapping reads and dividing by the library size (in millions of reads). The log2 normalized peak quantifications were used in further analyses. As a Q/C check, we used phantompeakqualtools[44] to calculate ChIP-seq cross-correlation statistics. We removed samples failing Q/C: those with total reads <10,000,000, read alignment fraction <75%, read duplication fraction >30%, normalized strand coefficient (NSC) <1.03, relative strand correlation (RSC) <0.5, or fraction of reads in peaks <11%. We also removed all cerebellum samples and samples originating from the Oxford or MRC London Brain Banks. This left us with a dataset of 56 ASD samples and 48 control samples from 35 ASD and 33 control brains.

**Similarity network fusion to identify molecular subtypes.** For 30 ASD and 17 control samples present in all four datasets, we used SNF[17] to cluster samples together based on their relationships across the four data types. Before running

SNF, we adjusted the datasets to remove the influence of technical and biological covariates. For mRNA expression, we fit a linear model for each gene: Expression ~ Diagnosis + Age + Sex + Region + RIN + Brain bank + Sequencing batch + seqStatPC1 + seqStatPC2 + seqStatPC3 + seqStatPC4 + seqStatPC5 and regressed out the effect of all covariates except Diagnosis. For miRNA expression, we fit a linear model for each miRNA transcript: Expression ~ Diagnosis + Age + Sex + Region + RIN + Brain bank + Proportion of reads mapping to exons + log10(Sequencing depth) + PMI and regressed out the effect of all covariates except Diagnosis. For DNA methylation, we fit a linear model for each gene promoter and gene body: Methylation ~ Diagnosis + Age + Sex + Region + Brain bank + Batch + CET and regressed out the effect of all covariates except Diagnosis. For histone acetylation, we fit a linear model for each H3K27ac peak: Acetylation ~ Diagnosis + Age + Sex + Region + Brain bank + CET + Fraction of reads in peaks + Duplicate read fraction + Aligned read fraction and regressed out the effect of all covariates except Diagnosis.

To identify ASD molecular subtypes, we restricted each dataset to its differential features between ASD and control samples. For mRNA expression, we restricted the genes to 2591 differentially expressed genes at an FDR < 10% from the idiopathic ASD vs control analysis of Parikshak et al.[9]. For miRNA expression, we restricted the miRNA transcripts to 92 differentially expressed miRNAs at an FDR < 10% from Wu et al.[12]. For DNA methylation, we ran an initial differential methylation analysis looking at all ASD vs control samples (for details, see below ASD vs control differential molecular analyses) and restricted the genes to 2578 differentially methylated promoters at an FDR < 10%. For histone acetylation, we ran an initial differential acetylation analysis looking at all ASD vs control samples (for details, see below ASD vs control differential molecular analyses) and restricted the peaks to 2156 differentially acetylated peaks at an FDR < 20%.

We ran SNF on the four adjusted and restricted datasets using the SNFtool package in R [https://cran.r-project.org/web/packages/SNFtool/index.html]. For each dataset, we normalized the values of each feature using the standardNormalization function in SNF. For each dataset, we calculated a sample-sample Euclidean distance using the dist2 function and used this distance to calculate a sample-sample affinity matrix using the affinityMatrix function with parameters: $K = 20$ and alpha = 0.5. We generated a fused affinity matrix combining all 4 affinity matrices using the SNF function with parameters: $K = 20$ and $T = 15$ and then used the spectralClustering function to demarcate the fused affinity matrix into 2 clusters of samples. Alternatively, we also used the symnmf_newton function in Matlab from the symNMF package[45] to cluster the fused affinity matrix.

**Classification of samples into the two SNF clusters.** For 61 ASD and 61 control samples that were missing from at least one of the four datasets, we built logistic regression models to classify them into one of the two clusters identified by SNF. For each of the four datasets, we calculated the sample loadings on the first

principal component (PC1) of its differential features used as input to SNF. These PC1 loadings were transformed into Z-scores and used as predictors in the models.

We built a total of twelve different logistic regression models. Three models were for samples present in three datasets: mRNA/miRNA/DNA methylation, mRNA/miRNA/histone acetylation, and mRNA/DNA methylation/histone acetylation. Six models were for samples present in two datasets: mRNA/miRNA, mRNA/DNA methylation, mRNA/histone acetylation, miRNA/DNA methylation, miRNA/histone acetylation, and DNA methylation/histone acetylation. Finally, three models were for samples present in one dataset: mRNA, DNA methylation, and histone acetylation. There were no samples that were present in miRNA/DNA methylation/histone acetylation or miRNA only datasets.

For each model, the training set was the 47 sample assignments identified by SNF. In training the model, the response variable was either 0 (for SNF group 1) or 1 (for SNF group 2) and the predictors were the sample differential Z-scores. We performed exhaustive leave one out validation on the training set by leaving each sample out, training the model with the remaining samples and predicting the response of the held-out sample. For each model, we chose a cutoff to distinguish between the two groups that maximized the cross-validation accuracy. We classified the test samples by running them through the model and applying the chosen cutoff.

**Outlier removal**. Before running differential analyses and Co-expression/co-methylation network analysis (WGCNA), we removed outlier samples from each of the four datasets. For each dataset, we calculated sample-sample correlations and removed samples with a signed Z-score >3[46].

**ASD vs control differential molecular analyses**. For mRNA expression, we ran differential expression analysis by fitting a linear mixed effect model for each gene: Expression ~ Diagnosis + Age + Sex + Region + RIN + Brain bank + Sequencing batch + seqStatPC1 + seqStatPC2 + seqStatPC3 + seqStatPC4 + seqStatPC5 as fixed effects, and brainID as a random effect. For miRNA expression, we ran differential expression analysis by fitting a linear mixed effect model for each miRNA transcript: Expression ~ Diagnosis + Age + Sex + Region + RIN + Brain bank + Proportion of reads mapping to exons + log10(Sequencing depth) + PMI as fixed effects, and brainID as a random effect. For DNA methylation, we ran differential methylation analysis by fitting a linear mixed effect model for each promoter and each gene body: Methylation ~ Diagnosis + Age + Sex + Region + Brain bank + Batch + CET as fixed effects, and brainID as a random effect. For histone acetylation, we ran differential acetylation analysis by fitting a linear mixed effect model for each H3K27ac peak: Acetylation ~ Diagnosis + Age + Sex + Region + Brain bank + CET + Fraction of reads in peaks + Duplicate read fraction + Aligned read fraction as fixed effects, and brainID as a random effect. For all differential analyses, we ran them separately for ASD Convergent subtype vs control samples and ASD Disparate subtype vs control samples. For DNA methylation and histone acetylation, we also ran an initial analysis looking at all ASD vs control samples before running SNF and sample classification. We used the R package nlme to fit linear mixed models [https://cran.r-project.org/web/packages/nlme/index.html].

**Co-expression/co-methylation network analysis**. For mRNA and miRNA expression, we used the co-expression networks defined in the previous studies[9,12]. To identify ASD-associated modules, we fit a linear mixed effect model for each co-expression module. For mRNA modules: module eigengene ~ Diagnosis + Age + Sex + Region + RIN + Brain bank + Sequencing batch + seqStatPC1 + seqStatPC2 + seqStatPC3 + seqStatPC4 + seqStatPC5 as fixed effects, and brainID as a random effect. For miRNA modules: module eigengene ~ Diagnosis + Age + Sex + Region + RIN + Brain bank + Proportion of reads mapping to exons + log10 (Sequencing depth) + PMI as fixed effects, and brainID as a random effect. We ran these analyses separately for ASD Convergent subtype vs control samples and ASD Disparate subtype vs control samples.

For DNA methylation, we generated co-methylation networks separately for both gene promoters and gene bodies. First, we set a linear model for each gene promoter and gene body as: Methylation ~ Diagnosis + Age + Sex + Region + Brain bank + Batch + CET and regressed out the effect of Brain bank. We generated networks with robust consensus WGCNA (rWGCNA)[47] using the WGCNA package in R[48]. We used a soft threshold power of 9 for gene promoters and 8 for gene bodies. We created 100 topological overlap matrices (TOMs) using 100 independent bootstraps of the samples with parameters: type = signed and corFnc = bicor. The 100 TOMs were combined edge-wise by taking the median of each edge across all bootstraps. The consensus TOM was clustered hierarchically using average linkage hierarchical clustering (using 1 − TOM as a dissimilarity measure). The topological overlap dendrogram was used to define modules using the cutreeHybrid() function with parameters: mms = 100, ds = 4, merge threshold of 0.1, and negative pamStage. To identify ASD-associated modules, we fit a linear mixed effect model for each co-methylation module: module eigengene ~ Diagnosis + Age + Sex + Region + Brain bank + Batch + CET as fixed effects, and brainID as a random effect. We ran these analyses separately for ASD Convergent subtype vs control samples and ASD Disparate subtype vs control samples.

**Prediction of miRNA target genes**. We predicted mRNA target genes for each miRNA using TargetScan v7.2[27]. We downloaded 3' UTR sequences of human genes and miRNA family information from the TargetScan database. For miRNAs that were identified in the previous publication[12] and not present in the TargetScan default predictions, we manually curated their family conservation by visually inspecting the multiz 46-way vertebrate alignment at their genomic locus in the hg19 human assembly of the UCSC genome browser[49]. For each putative miRNA-UTR target site, TargetScan calculates a context++ score which takes into account both evolutionary conservation and targeting efficiency. These context++ scores were weighted based on affected isoform ratios. We took the top weighted context ++ score for each unique miRNA-UTR target pair.

To assess enrichment of targets within miRNA co-expression modules, we first filtered for the top 25% miRNAs by connectivity (module hubs) within each module (kME ≥0.84, 0.82, and 0.57 for the brown, magenta, and yellow modules, respectively) and identified the strongest targets with a context++ score ≤−0.05 for these hub genes.

**Assignment of H3K27ac regions to cognate gene**. We assigned H3K27ac regions within promoter regions (2KB upstream of TSS to TSS) to their proximal gene. For H3K27ac regions that did not lie within a gene promoter, we assigned them to their cognate gene using adult brain eQTL data and Hi-C data from bulk adult brain tissue[31] as well as Hi-C data from sorted NeuN+ and NeuN- cells from adult brain tissue (Synapse accession number: syn10248174 for NeuN-, syn10248215 for NeuN+). For eQTL data, we assigned a H3K27ac region to a gene if the eSNP resided within the H3K27ac peak. For Hi-C data, we assigned a H3K27ac region to a gene if the promoter of that gene physically interacted with a region containing the H3K27ac peak at an FDR < 1%. We assessed all possible H3K27ac region to gene linkages even if they were not consistent between eQTL and Hi-C datasets.

**Enrichment analyses**. We downloaded post-mortem brain single nucleus gene expression data from Hodge et al.[22]. The count data were normalized using log2 (CPM + 1). To identify markers of neuronal cell types, we ran differential expression analyses for a particular cell cluster against all other clusters when restricting the dataset to inhibitory neurons or excitatory neurons separately. Differential expression analyses were run in R using a linear model: expression ~ cluster membership. For each cluster, we identified markers as those genes with an FDR corrected P-value < 0.05 and a log2(fold change) >0.75.

We downloaded cell type markers for neurons, astrocytes, oligodendrocytes, endothelial cells, and microglia from Zhang et al.[50]. We downloaded microglial cell type-specific markers (fold change ≥1) from Hammond et al.[25]. We downloaded markers of microglial activation from Hirbec et al.[26]. For module cell type enrichments, enrichments of co-expression vs co-methylation modules, and enrichment with orthogonal gene lists we used logistic regression to test whether gene set 1 ~ gene set 2 using a background set of genes shared between study 1 and study 2.

For expression and methylation gene ontology enrichments, we used the g:Profiler[51] package in R with parameters: correction_method = fdr, max_set_size = 1000, and hier_filtering = moderate. We performed ordered queries with genes ordered by fold change for differential expression and methylation or by connectivity to the module eigengene (kME) for co-expression and co-methylation modules. For acetylation gene ontology enrichments, we used GREAT[28] [http://great.stanford.edu/public/html/index.php] with the default basal plus extension association rule setting.

**Partitioned heritability**. We ran stratified LD-score regression[33] to test for enrichment of common variant heritability from GWAS studies of ASD[4,34], Alzheimer's disease[36], and Inflammatory bowel disease[35] in genomic regions of interest. We downloaded the full baseline model of 53 functional categories [https://github.com/bulik/ldsc/wiki/Partitioned-Heritability] and included them with each calculation of partitioned heritability. For differentially expressed, differentially methylated, and co-expression/co-methylation modules, we defined their genomic regions as each gene body ±10 KB. For differentially acetylated regions, we defined the genomic region as each H3K27ac peak ±1 KB.

**Comparison to transcriptomic signatures of other disorders**. We corrected mRNA expression data from ASD Disparate subtype and control samples for technical and biological covariates by fitting a linear model for each gene: Expression ~ Diagnosis + Age + Sex + Region + RIN + Brain bank + Sequencing batch + seqStatPC1 + seqStatPC2 + seqStatPC3 + seqStatPC4 + seqStatPC5 and regressing out the effect of all covariates except Diagnosis. We downloaded differentially expressed genes at an FDR < 5% for ASD, Schizophrenia, Bipolar disorder, Major depressive disorder, and Alcoholism from a cross-disorder analysis of neuropsychiatric disorders[21]. For each disorder, we calculated the first principal component on the corrected expression when restricting to genes differentially expressed in that disorder. We checked for differential loading between ASD Disparate subtype samples and control samples using a two-sided Mann–Whitney U test.

**Assessment of transcriptome in other cortical regions**. We previously sequenced 87 samples in 4 additional cortical regions (BA4-6, BA7, BA17, BA38) from the individuals in this study[21] (Synapse accession number syn11242290). We mapped sequencing reads onto the hg19 genome using STAR[52] and calculated RNA-seq quality control metrics using PicardTools [http://broadinstitute.github.io/picard/]. We quantified gene expression using RSEM[53] with gencode v25 annotations[42]. We corrected the expression data for technical and biological covariates by fitting a linear mixed model for each gene: Expression ~ Region + Batch + Age + Sex + Diagnosis + Ancestry_Genotype + PMI + RIN + picard_rnaseq. PCT_CORRECT_STRAND_READS + picard_rnaseq.PCT_MRNA_BASES + picard_gcbias.AT_DROPOUT + star.multimapped_percent + picard_alignment. PCT_CHIMERAS + star.multimapped_toomany_percent + picard_insert.MEDIAN_INSERT_SIZE + picard_rnaseq.PCT_INTERGENIC_BASES + picard_rnaseq.MEDIAN_5PRIME_BIAS + picard_rnaseq.PCT_UTR_BASES + star.num_ATAC_splices + star.num_GCAG_splices + star.num_splices + star.avg_mapped_read_length + Age_sqd + picard_alignment.PCT_PF_READS_ALIGNED_sqd + picard_rnaseq.PCT_CORRECT_STRAND_READS_sqd + star.avg_mapped_read_length + star.num_ATAC_splices_sqd + star.num_annotated_splices_sqd + star.num_GCAG_splices_sqd as fixed effects and subject as a random effect. We regressed out the effect of all technical covariates, which created an expression dataset containing the effects of only biological covariates (subject, diagnosis, region, age, age squared, sex, and ethnicity).

For each region, we calculated the sample loadings across the first principal component (PC1) of gene expression for 2591 differentially expressed genes at an FDR < 10% from the idiopathic ASD vs control analysis of Parikshak et al.[9]. These PC1 loadings were then transformed into Z-scores.

**Reporting summary**. Further information on research design is available in the Nature Research Reporting Summary linked to this article.

## Data availability

Full range of values underlying heatmaps in Figs. 2f, i, 4e, f, 5a, b and Supplementary Figs. 7e, g, 8e, f, 9f, g, 10f, g, 11a–d, 13a–c are provided as a Source Data File. Harvard Autism Tissue Program [https://hbtrc.mclean.harvard.edu/], NIH Neuro Brain Bank [http://www.medschool.umaryland.edu/btbank/], Oxford Brain Bank [https://www.hra.nhs.uk/planning-and-improving-research/application-summaries/research-summaries/the-oxford-brain-bank/], MRC London Brain Bank [https://brainbanknetwork.cse.bris.ac.uk/], Raw data for mRNA, miRNA, DNA methylation, and H3K27ac from ASD and control brains (Synapse.org accession number syn4587609), Gencode [https://www.gencodegenes.org/], hg19 genome [http://genome.ucsc.edu/], TargetScan DB [http://www.targetscan.org/cgi-bin/targetscan/data_download.vert72.cgi], Psychencode eQTL and Hi-C datasets [http://resource.psychencode.org/] (Synapse.org accession number: syn10248174 for NeuN-, syn10248215 for NeuN+), LD-score regression model [https://github.com/bulik/ldsc/wiki/Partitioned-Heritability], Gene expression of ASD and control brains from other cortical regions (Synapse.org accession number syn11242290). All other relevant data supporting the key findings of this study are available within the article and its Supplementary Information files or from the corresponding author upon reasonable request. A reporting summary for this article is available as a Supplementary Information file. Source data are provided with this paper.

## Code availability

Underlying R code to run SNF clustering and ASD/Control differential analyses is available at [https://github.com/dhglab/ASD-Integration-Subtypes-Manuscript].

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

## Acknowledgements

We thank Neelroop Parikshak and Ye Emily Wu for assistance in obtaining datasets, William Pembroke and Damon Polioudakis for assistance with figures, as well as members of the Geschwind lab for stimulating discussions. Funding for this work was provided by grants to D.H.G (NIMH R01MH110927, U01MH115746 and R01MH094714), G.R. (NIMH 1F32MH114620), H.W. (NIMH R00MH113823 and NARSAD Young Investigator Award), M.J.G. (SFARI Bridge to Independence Award, NIMH R01MH121521, and NIMH R01MH123922), S.P. (Core funds from A*STAR Singapore), and J.M. (Medical Research Council R005176 and K013807).

## Author contributions

G.R. and D.H.G. planned the analyses and wrote the paper with assistance from all authors. The analyses were performed by G.R. with help from H.W., M.J.G., J.H. and J.C.W. H.W. processed all Hi-C datasets. C.C.Y.W. and J.M. generated, normalized, and provided guidance on the DNA methylation dataset. W.S. and S.P. generated and provided guidance on the histone acetylation dataset.

## Competing interests

The authors declare no competing interests.
