## [Peer Review File · Nature Communications]

Reviewers' comments:

Reviewer #1 (Remarks to the Author):

Overall: This is a very interesting study that describes a novel application of a method, called Similarity Network Fusion (SNF), to integrate across multi-omics data on the autism brain using datasets from existing transcriptomic, DNA methylation, miRNA expression, and histone acetylation studies. Moreover, the study also uses the acetylome data coupled with existing GWAS data to link specific ASD-associated SNPs to gene expression based on eQTL and Hi-C analyses, thus identifying potential causal relationships between some of the SNP-containing acetylated regions and dysregulated gene expression. Importantly, the SNF analysis revealed two distinct molecular subtypes of ASD (convergent and disparate). This reduction in the molecular heterogeneity of the samples provided increased power to identify considerably more differentially expressed genes and methylated regions in the brain of convergent samples in comparison to those revealed in the original studies on the combined set of ASD samples. Interestingly, no mRNA and miRNA co-expression (or co-methylation) modules or differentially acetylated peaks were identified for the "disparate" samples. In summary, this is a much-needed integration of large-scale genome-wide data from multiple studies that converge on key processes disrupted in the frontal and temporal cortices in a subgroup of autistic individuals. It remains to be seen what the key underlying neuropathological processes are for the disparate subgroup of ASD individuals.

Comment

While there is ample data contained within the manuscript on the mRNA expression, DNA methylation, and histone acetylation analyses, there is relatively little included data (no figures or table) on the miRNA expression analysis. There is also no description of the overall functional significance of the mRNA modules containing targets of the ASD-significant miRNA modules. Given that there are only 43 differentially expressed miRNAs in the convergent subtype, it would be helpful to provide at least a table of these within the manuscript to allow ready comparisons with other miRNA expression studies in ASD. Alternatively, the 43 DE miRNAs could be highlighted in the Supplemental table that contains all miRNA expression values.

Reviewer #2 (Remarks to the Author):

The manuscript by Ramaswami et al. takes a multi-omic approach to characterize molecular subgroups of autism spectrum disorder (ASD) and uses those more homogeneous groupings to identify new DNA methylation, micro and messenger RNA expression, and histone acetylation changes related to each subgroup. This is all evaluated in postmortem frontal and temporal brain cortex tissue - the presumed affected tissue - and, thus, lends itself to informing potential mechanisms in ASD. The Authors utilized the Similarity Network Fusion (SNF) method and identified 2 molecular subgroups of ASD, termed "Convergent" and "Disparate". No significant differences in the epigenome or transcriptome were identified in the "disparate" group, however, several novel changes were found in the convergent group.

These results provide important new insights into ASD by defining molecular subgroups, for the first time, and by identifying biologic pathways that are altered in the convergent subgroup. Overall the manuscript was well written, the study design and methods were appropriate, and the conclusions were justified given the results. However, I have two minimal questions/points that are mainly meant to improve the manuscript.

1. The authors report enrichment of ASD, IBD, and Alzheimers genetic risk variants in the regulatory regions they identified using stratified LD score regression. It would be helpful to provide the rationale for why IBD and Alzheimers were specifically chosen for this analysis. For

example, is there reason to think common risk variants for those 2 diseases would or would not show enrichment for regulatory regions? What about for other brain disorders?

2. I'm not sure why DNA methylation in the gene body is expected to be negatively associated with expression (lines 283-284)? Previous studies have shown this is true for most promoter regions but typically the gene body has been shown to be positively correlated with expression (e.g. increased methylation and increased gene expression); see PMIDs: 19329998, 19829295, 19139413)

Reviewer #3 (Remarks to the Author):

The authors of the Ramaswami et al manuscript present the joint analysis of the four different assays (mRNA expression, miRNA expression, DNA methylation, and histone acetylation) over set of 48 autistic and 45 control human postmortem brains. Each of the individual assays have previously been generated and published by the same group or by close collaborators. The main contribution of the current manuscript is that the authors identify a subset of the samples from autistics brains that have similar functional profiles as measured through a combination of the four assays. The rest of the autism brains seem to exhibit diverse functional measurements and be more similar to the control brains. The authors then focus on re-analyzing the individual assays by contrasting the coherent autism brains and the control brains. This approach enables them to improve upon the result from the prior publications. For example, they identify substantially more differentially expressed genes and gene components and to associate these better with particular brain cell types. Finally, they identify enrichment of common-variants-based heritability in differentially acetylated regions associated with neuronal genes. The numerous observations are summarized by a model in which DNA variants, microRNA, DNA methylation, and histone acetylation dysregulation directly or indirectly downregulate neuronal genes and upregulate immune-glial genes.

Everything (text, figures, method description, etc.) in the paper is high quality. The topic of transcriptomic and epigenomic dysregulation in autism brains is of high interest to a large scientific community. The group has a very strong record in the field is the right team to tackle the non-trivial problem of integrating the large number of diverse datasets. I am not able to evaluate every part of the paper—there are a lot of datasets and methods used that I do not know well. But I was not able to find anything weak in the places I looked more carefully. I would enthusiastically support the publication of the manuscript.

Summary

We thank all 3 reviewers who provided a very positive assessment of the manuscript, appreciating the novel application of SNF to ASD datasets and praised our integrative approach to characterize the molecular etiology of ASD. The importance and novelty of the work is highlighted by some of the summary comments from reviewers: *"In summary, this is a much-needed integration of large-scale genome-wide data from multiple studies that converge on key processes disrupted in ASD (reviewer 1);" "These results provide important new insights into ASD by defining molecular subgroups, for the first time... (reviewer 2);" "Everything (text, figures, method description, etc.) in the paper is high quality. The topic of transcriptomic and epigenomic dysregulation in autism brains is of high interest to a large scientific community. The group has a very strong record in the field is the right team to tackle the non-trivial problem of integrating the large number of diverse datasets.... I would enthusiastically support the publication of the manuscript. (reviewer 3)"*

Comments from reviewers 1 & 2 identify a few minor areas of improvement that we have completely addressed in our revised manuscript. Please find a point by point response to the reviewers' comments below.

Response to Reviewers' comments

Reviewer #1 (Remarks to the Author):

Overall: This is a very interesting study that describes a novel application of a method, called Similarity Network Fusion (SNF), to integrate across multi-omics data on the autism brain using datasets from existing transcriptomic, DNA methylation, miRNA expression, and histone acetylation studies. Moreover, the study also uses the acetylome data coupled with existing GWAS data to link specific ASD-associated SNPs to gene expression based on eQTL and Hi-C analyses, thus identifying potential causal relationships between some of the SNP-containing acetylated regions and dysregulated gene expression. Importantly, the SNF analysis revealed two distinct molecular subtypes of ASD (convergent and disparate). This reduction in the molecular heterogeneity of the samples provided increased power to identify considerably more differentially expressed genes and methylated regions in the brain of convergent samples in comparison to those revealed in the original studies on the combined set of ASD samples. Interestingly, no mRNA and miRNA co-expression (or co-methylation) modules or differentially acetylated peaks were identified for the "disparate" samples. In summary, this is a much-needed integration of large-scale genome-wide data from multiple studies that converge on key processes disrupted in the frontal and temporal cortices in a subgroup of autistic individuals. It remains to be seen what the key underlying neuropathological processes are for the disparate subgroup of ASD individuals.

Comment

While there is ample data contained within the manuscript on the mRNA expression, DNA methylation, and histone acetylation analyses, there is relatively little included data (no figures or table) on the miRNA expression analysis. There is also no description of the overall functional significance of the mRNA modules containing targets of the ASD-significant miRNA modules. Given that there are only 43 differentially expressed miRNAs in the convergent subtype, it would be helpful to provide at least a table of these within the manuscript to allow ready comparisons with other miRNA expression studies

in ASD. Alternatively, the 43 DE miRNAs could be highlighted in the Supplemental table that contains all miRNA expression values.

We thank the reviewer for their positive comments on the manuscript and for the suggestion to expand upon the miRNA analysis results. The identification of differentially expressed miRNA transcripts and co-expression modules in ASD as reported in our previous work (Wu et al. Nature Neuroscience 2016) was largely recapitulated in this study. We note that DNA methylation, and histone acetylation analyses were substantially more powered when stratifying by subtype, thus we focused the main text and figures on these three moieties. We believe this disparity in power reflects the feature size disparity between the 4 datasets. There are only 800 total miRNA transcripts, and ASD sample heterogeneity does not seem to hinder our ability to identify differential transcripts. In contrast the other 3 datasets have between 15000-50000 features, a 20-60 fold increase over miRNA, and are therefore more sensitive to sample heterogeneity.

In response to this reviewer's comment, we have annotated Supplementary Table 3 to highlight the 43 DE miRNAs including the 10 newly associated with ASD, and mention the high overlap with the previous analysis in the main text. We do agree with the reviewer that additional description of the functional significance of mRNA co-expression modules enriched with ASD-associated miRNA targets is warranted and have now added into the main text:

"We found an enrichment of genes in the upregulated mRNA.M19 module within the predicted targets of the miRNA.brown module (**Supplementary Figure 8f**), suggesting the downregulation of these miRNAs may contribute to the upregulation of immune processes in ASD. We also found an enrichment of genes in the downregulated mRNA.M16 and mRNA.M3 modules within the predicted targets of miRNA.magenta and miRNA.yellow (**Supplementary Figure 8f**), suggesting the upregulation of these miRNAs may contribute to the downregulation of neuronal processes in ASD."

Reviewer #2 (Remarks to the Author):

The manuscript by Ramaswami et al. takes a multi-omic approach to characterize molecular subgroups of autism spectrum disorder (ASD) and uses those more homogeneous groupings to identify new DNA methylation, micro and messenger RNA expression, and histone acetylation changes related to each subgroup. This is all evaluated in postmortem frontal and temporal brain cortex tissue - the presumed affected tissue – and, thus, lends itself to informing potential mechanisms in ASD. The Authors utilized the Similarity Network Fusion (SNF) method and identified 2 molecular subgroups of ASD, termed "Convergent" and "Disparate". No significant differences in the epigenome or transcriptome were identified in the "disparate" group, however, several novel changes were found in the convergent group.

These results provide important new insights into ASD by defining molecular subgroups, for the first time, and by identifying biologic pathways that are altered in the convergent subgroup. Overall the manuscript was well written, the study design and methods were appropriate, and the conclusions were justified given the results. However, I have two minimal questions/points that are mainly meant to improve the manuscript.

We thank the reviewer for their positive comments on the manuscript address their two comments below.

1. The authors report enrichment of ASD, IBD, and Alzheimers genetic risk variants in the regulatory regions they identified using stratified LD score regression. It would be helpful to provide the rationale for why IBD and Alzheimers were specifically chosen for this analysis. For example, is there reason to think common risk variants for those 2 diseases would or would not show enrichment for regulatory (sic) regions? What about for other brain disorders?

The reviewer asks is there a reason to think that common risk variants for the two comparison diseases would show enrichment in regulatory regions. Yes, in general, all common diseases, including those above, partition the majority of their genetic risk into regulatory regions (typically > 70%+). We chose IBD and Alzheimer's disease as comparator datasets for GWAS enrichments because previous work has established that immune-related genes are upregulated in ASD, but are not enriched in causal genetic variation. Both IBD and AD genetic risk directly implicate immune genes and pathways, albeit distinctly. Thus, an expectation was that IBD would be somewhat of a "positive control" dataset to verify that the partitioned heritability method was generating the expected result of an enrichment of ASD-upregulated genes with IBD risk-variants. We chose Alzheimer's disease as a "negative control" dataset, because it is a disorder that also impacts the brain and has immune related risk variation, but has a distinct genetic risk profile. The Alzheimer's disease GWAS is also of similar sample size to the ASD GWAS (17,000 vs 18,000 cases), and thus is similarly powered. We have modified the main text to include the rationale behind choosing these two GWAS datasets:

"We included Inflammatory Bowel Disease (IBD) as comparator GWAS dataset, because IBD is a disorder of immune dysregulation, but does not affect the brain. We also included Alzheimer's Disease (AD), because AD is a neurological disorder that involves some neural immune pathways, but has a distinct genetic profile to ASD."

2. I'm not sure why DNA methylation in the gene body is expected to be negatively associated with expression (lines 283-284)? Previous studies have shown this is true for most promoter regions but typically the gene body has been shown to be positively correlated with expression (e.g. increased methylation and increased gene expression); see PMIDs: 19329998, 19829295, 19139413)

We agree with the reviewer that previous studies assessing the relationship between gene body methylation and gene expression have been mixed, suggesting both negative and positive correlations. In ASD datasets, gene body methylation is negatively correlated with expression, but this relationship is not universal across all contexts. We have modified the main text:

"As expected, there was a negative correlation between differential expression and differential methylation for gene promoters (**Figure 3c**). Surprisingly, there was also a negative correlation with gene body methylation (**Figure 3d**), suggesting that in the context of ASD, gene body methylation is associated with negative regulation of gene expression."

Reviewer #3 (Remarks to the Author):

The authors of the Ramaswami et al manuscript present the joint analysis of the four different assays (mRNA expression, miRNA expression, DNA methylation, and histone acetylation) over set of 48 autistic and 45 control human postmortem brains. Each of the individual assays have previously been generated and published by the same group or

by close collaborators. The main contribution of the current manuscript is that the authors identify a subset of the samples from autistic brains that have similar functional profiles as measured through a combination of the four assays. The rest of the autism brains seem to exhibit diverse functional measurements and be more similar to the control brains. The authors then focus on re-analyzing the individual assays by contrasting the coherent autism brains and the control brains. This approach enables them to improve upon the result from the prior publications. For example, they identify substantially more differentially expressed genes and gene components and to associate these better with particular brain cell types. Finally, they identify enrichment of common-variants-based heritability in differentially acetylated regions associated with neuronal genes. The numerous observations are summarized by a model in which DNA variants, microRNA, DNA methylation, and histone acetylation dysregulation directly or indirectly downregulate neuronal genes and upregulate immune-glia genes.

Everything (text, figures, method description, etc.) in the paper is high quality. The topic of transcriptomic and epigenomic dysregulation in autism brains is of high interest to a large scientific community. The group has a very strong record in the field is the right team to tackle the non-trivial problem of integrating the large number of diverse datasets. I am not able to evaluate every part of the paper—there are a lot of datasets and methods used that I do not know well. But I was not able to find anything weak in the places I looked more carefully. I would enthusiastically support the publication of the manuscript.

We thank the reviewer for their enthusiastic comments on the manuscript.